# Evaluating the impact of the Bolsa Familia conditional cash transfer program on premature cardiovascular and all-cause mortality using the 100 million Brazilian cohort: a natural experiment study protocol

Julia M Pescarini [1,2] Peter Craig,[3] Mirjam Allik,[3] Leila Amorim,[4] Sanni Ali,[1,5] Liam Smeeth,[5,6] Mauricio L Barreto,[1,7] Alastair H Leyland,[3] Estela M L Aquino,[1,7] Srinivasa Vittal Katikireddi [3]

For numbered affiliations see end of article.

**Correspondence to**
Dr Julia M Pescarini;
juliapescarini@gmail.com

## ABSTRACT

**Introduction** Brazil's Bolsa Familia Program (BFP) is the world's largest conditional cash transfer scheme. We shall use a large cohort of applicants for different social programmes to evaluate the effect of BFP receipt on premature all-cause and cardiovascular mortality.

**Methods and analysis** We will identify BFP recipients and non-recipients among new applicants from 2004 to 2015 in the 100 Million Brazilian Cohort, a database of 114 million individuals containing sociodemographic and mortality information of applicants to any Brazilian social programme. For individuals applying from 2011, when we have better recorded income data, we shall compare premature (age 30–69) cardiovascular and all-cause mortality among BFP recipients and non-recipients using regression discontinuity design (RDD) with household monthly per capita income as the forcing variable. Effects will be estimated using survival models accounting for individuals follow-up. To test the sensitivity of our findings, we will estimate models with different bandwidths, include potential confounders as covariates in the survival models, and restrict our data to locations with the most reliable data. In addition, we will estimate the effect of BFP on studied outcomes using propensity score risk-set matching, separately for individuals that applied ≤2010 and >2011, allowing comparability with RDD. Analyses will be stratified by geographical region, gender, race/ethnicity and socioeconomic position. We will investigate differential impacts of BFP and the presence of effect modification for a combination of characteristics, including gender and race/ethnicity.

**Ethics and dissemination** The study was approved by the ethics committees of Oswaldo Cruz Foundation and the University of Glasgow College of Medicine and Veterinary Life Sciences. The deidentified dataset will be provided to researchers, and data analysis will be performed in a safe computational environment without internet access. Study findings will be published in high quality peer-reviewed research articles. The published results will be disseminated in the social media and to policy-makers.

## Strengths and limitations of this study

► Few previous studies of conditional cash-transfer programmes have investigated impacts on adult health, or on premature death by cardiovascular diseases, using individual-level data on exposures and outcomes.

► We exploit a nationwide linkage of social policy and health datasets to evaluate the largest conditional cash transfer in the world in one of the most unequal countries.

► We use natural experimental approaches to estimate the effect of Bolsa Familia Program that control for both observed and unobserved differences between recipients and non-recipients.

► Limitations associated with the use of routinely collected data include underascertainment of deaths, imperfect measurement of incomes, and a restricted range of covariates.

► The period of follow-up is limited to 10 years, so may be insufficient to observe long-term impacts, including life course effects of improving socioeconomic conditions in childhood.

## INTRODUCTION

Conditional cash transfer (CCT) programmes have been widely implemented since the 1990s, aiming to reduce poverty among groups largely excluded from previous social policies.[1] A World Bank Study using data from 79 countries suggested that such programmes have reduced absolute poverty (people living with ≤US$1.90 per capita per day) by 36% and relative poverty (the poorest 20% in each country) by 8%.[2] CCT programmes impose additional requirements on recipients, most commonly a health and/

or education component that targets children and pregnant/breastfeeding women.[1] Therefore, the evidence of the impact of cash transfer programmes on health derives largely (but not exclusively) from studies evaluating the impact on food availability, nutrition, child and maternal health.[3–5]

In low-income and middle-income countries, cardiovascular diseases (CVD) still lead as the number one cause of death among non-communicable diseases (NCD), accounting for 9.6 million deaths in 2017, but mortality rates have been mainly increasing in middle-income countries.[6 7] A recent systematic review notes that low socioeconomic status, high alcohol intake, obesity, diabetes, hypertension, physical inactivity and smoking are the main modifiable factors associated with early mortality due to CVD.[8] Premature mortality (ie, death among persons 30–69 years of age) is an important indicator included in the sustainable developmental goals 3.4 target[9] for monitoring the implementation of effective public policies for disease prevention and control.[10]

In Brazil, poverty is largely concentrated among women, blacks and individuals living in rural areas.[11] The epidemiological transition in the country occurred heterogeneously, and the overall increase in life expectancy was accompanied by a greater decrease in CVD mortality in regions with better socioeconomic conditions.[12 13] In 2013, a nationwide survey highlighted a higher prevalence of risk factors for CVD (ie, smoking, heavy alcohol use, physical inactivity and sedentary lifestyle, as well as the lower consumption of fruits and vegetables) among Brazilians from non-white ethnic backgrounds and those with lower education, illustrating the relationship between poverty and racial/ethnic, gender and income disparities in health.[14]

The Bolsa Familia Program (BFP), the world's largest CCT, benefits over 13 million families a year and has helped to reduce poverty and inequality in Brazil.[15–17] Although the programme has had large effects on child and maternal mortality, its impact on NCDs, especially on cardiovascular deaths, remains unknown. Therefore, the aims of our study are to investigate the effects of the BFP on CVD (ischaemic heart disease and cerebrovascular disease) mortality and all-cause mortality, and to assess how these effects differ by socioeconomic position, race/ethnicity, urbanicity and region.

Our detailed objectives are:
1. To estimate the causal effect of BFP on premature cardiovascular disease mortality.
2. To estimate the causal effect of BFP on premature ischaemic heart disease and cerebrovascular disease mortality.
3. To estimate the causal effect of BFP on all-cause premature mortality.
4. To investigate whether the causal effects of BFP on cardiovascular and all-cause mortality differ by population subgroups, including gender, educational attainment, race/ethnicity, geographical region, urbanicity and socioeconomic position.
5. To explore how combinations of selected social characteristics influence the causal effects of BFP on the above outcomes, adopting an intersectionality perspective.

## METHODS AND ANALYSIS
### Study design
This study will be analysed as a retrospective, dynamic and open cohort, linking data from the baseline registries of individuals in the 100 Million Brazilian Cohort, with data on BFP intervention receipt and individual-level mortality records for the whole country.

### Intervention
We report key implementation characteristics as per the TIDieR-PHP template.[18] BFP was implemented in 2004 and involves cash payments to poor families within Brazil, conditional on educational and health requirements.[16] Under BFP, cash benefits are preferentially paid to women in qualifying households. To be eligible, households must be registered in the Brazil's National Registry for Social Programmes 'Cadastro Único' (CadUnico), and have a household income below a defined extreme poverty threshold (monthly per capita household income ≤BRL77 in 2014 (approximately US$19)) or poverty threshold (monthly per capita household income ≤BRL154 in 2014 (approximately US$39)). A number of changes to income thresholds for eligibility to BFP have occurred

**Table 1** Eligibility criteria for Bolsa Familia Program receipt in Brazil between 2004 and 2015 and income standardisation rates

| Year | Extremely poverty eligibility criteria* | Poverty eligibility criteria* | Date of change | Income standardisation rate† from 2004 to 2014 thresholds |
|---|---|---|---|---|
| 2004 | ≤BRL 50 (US$12.5) | ≤BRL 100 (US$26) | | Income × 1.540 |
| 2007 | ≤BRL 60 (US$15) | ≤BRL 120 (US$30) | 28 December 2007 | Income × 1.283 |
| 2009 | ≤BRL 70 (US$17.5) | ≤BRL 140 (US$35) | 1 September 2009 | Income × 1.100 |
| 2014 | ≤BRL 77 (US$19.3) | ≤BRL 154 (US$38.5) | 1 June 2014 | Income |

*Values in USD were calculated based on the following exchange rate: 4BRL=US$1.
†For each year (2004, 2007 and 2009), the standardised rate was estimated by dividing the poverty (or extreme poverty) criteria from 2014 by the poverty criteria (or extreme poverty) from that year.

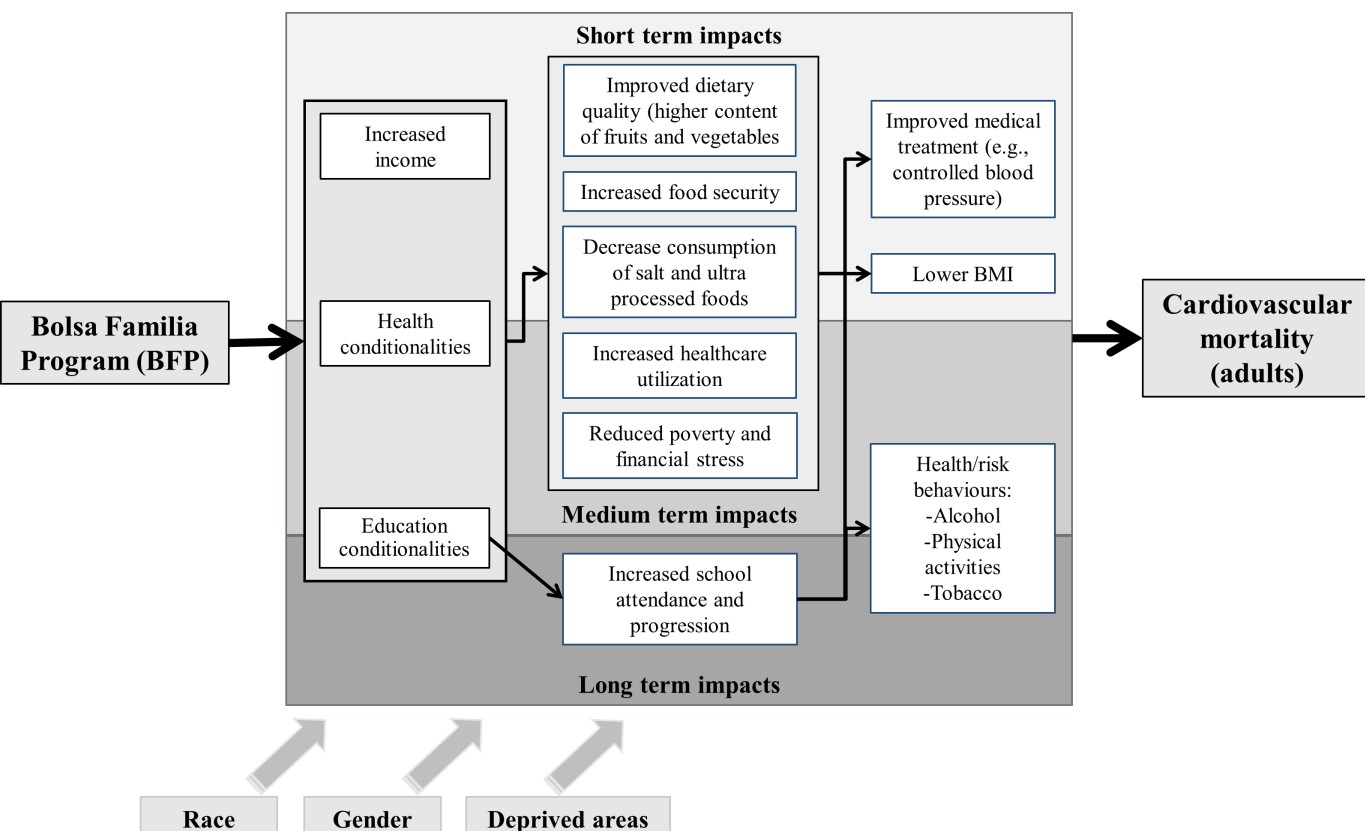

**Figure 1** Logic model of the potential effect of Bolsa Familia Program (BFP) on all-cause and cardiovascular mortality. BMI, body mass index.

from 2004 onwards (table 1). Extremely poor families receive a fixed benefit of BRL77 plus additional amounts for pregnant women, children and adolescents. Families defined as poor receive the supplements for pregnant women, children and any adolescents in the household, but not the fixed benefit. In the first years of BFP implementation, indigenous and quilombola communities (old African settlements) were prioritised to start receiving the benefit.[16] Receipt of benefit is conditional on families meeting certain qualifying conditions: pregnant women must access prenatal care; children aged 0–5 years and breastfeeding women must undergo monitoring by health professionals (vaccination and health check-ups at home or in the nearest primary healthcare centre); and school-age children must attend school for at least 85% of school days. If individuals no longer meet the inclusion criteria, that is, if they improve their socioeconomic status, do not meet the conditionalities or do not update the registry every 2 years, the benefit will only continue for two more years. Nevertheless, non-compliant families are thought to be more vulnerable and, in these cases, receive a visit of a social worker that will help families' compliance and their maintenance in the programme.[19]

## Logic model

We created a logic model (figure 1), informed by the existing literature, to describe the hypothesised mechanisms through which BFP may have an effect on CVD outcomes. We identified pathways that are likely to operate through material impacts, education and health conditionalities, and by fostering social inclusion, as well as distinguishing short, medium and long-term timescales. Possible short-term impacts include changes in nutritional status in adults, through reduced salt and ultraprocessed foods intake, increased consumption of foods with high nutrient content, and increases in overall energy intake.[20–25] BFP can also affect the socioeconomic conditions of the household, including more working hours, better jobs and higher income.[26–28] Effects of CCTs that may be observed at in the medium term include those intermediated by the health conditionalities of BFP, such as increased healthcare utilisation.[29–31] Possible long-term impacts include those stimulated by reductions in early life and cumulative socioeconomic adversity,[32 33] improved access to education and upward social mobility.[34 35]

Other potential impacts include the cumulative effects of better nutrition, reduced chronic stress and improved access to primary health services,[21 36–38] which can contribute to the control of hypertension, obesity and diabetes and longer survival following acute CVD events.[39–41] We hypothesise that inclusion of BFP beneficiary families in the Family Health Program might promote early CVD diagnosis and better care,[42] even though Brazil has a Universal Healthcare System (SUS) and access to free hypertension drugs has substantially increased over recent decades.[43] Also, by providing an

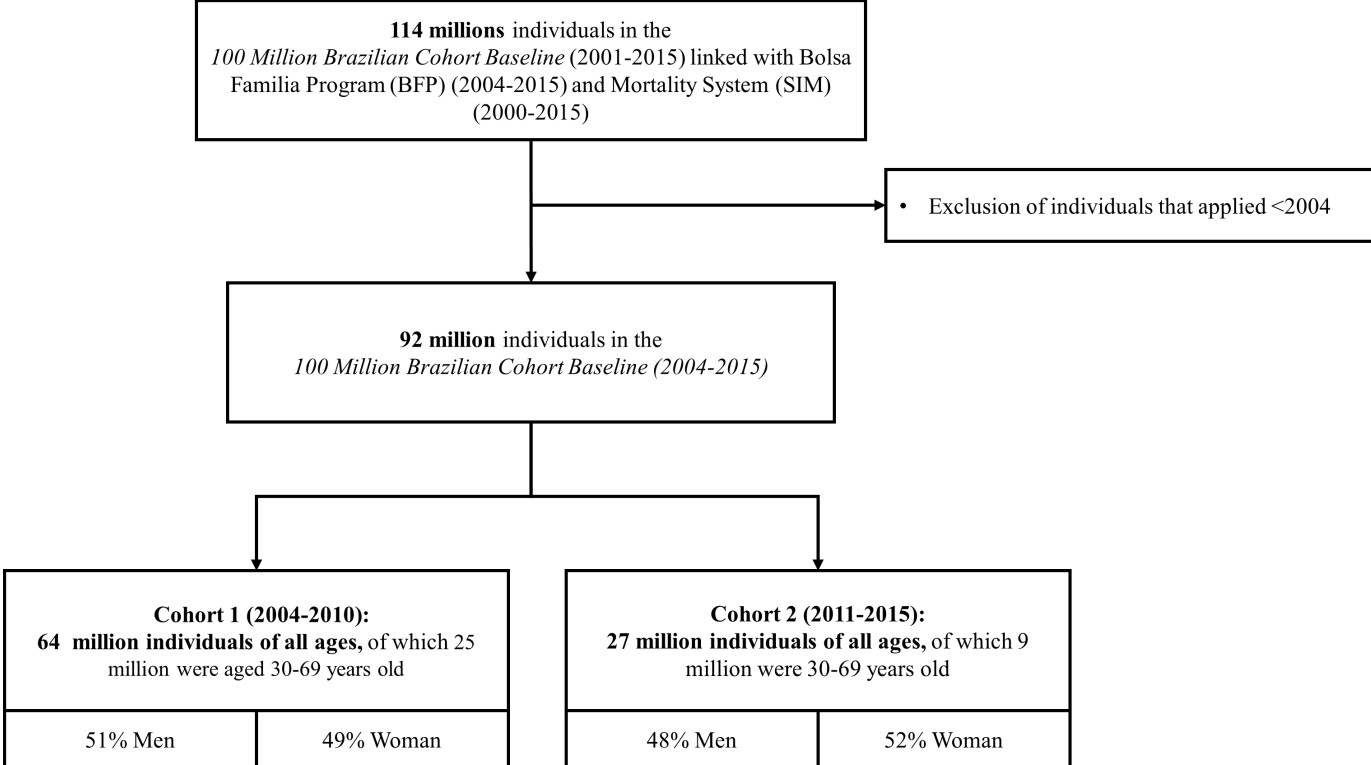

**Figure 2** Flow chart of data selection.

## Datasets

Our sample comprises members of families who applied for BFP or other social benefits from 2001 to 2015 registered in CadÚnico, combined as part of the baseline of *The 100 Million Brazilian Cohort*.[50] We deterministically linked the baseline information of individuals from the 100 Million Brazilian Cohort (2001–2015) with BFP data (2004–2015) and used a semideterministic linkage based on five identifiers (name, sex, year of birth, name of the mother and municipality of residency) to link with individual-level mortality records from the Mortality Information System (2001–2015).[50] From 2001 to 2015, over 114 million individuals applied for social benefits, but the final deidentified dataset will include information on the 92 million individuals that applied between 2004 and 2015 (figure 2).

### Sociodemographic variables

The baseline of the 100 Million Brazilian Cohort includes a range of sociodemographic variables collected at individuals' first application in CadÚnico, and includes household income, gender, age, race/ethnicity, geographical region and urban–rural classification, housing characteristics and education (table 2). As the cohort was built from different versions of CadÚnico (ie, V.6 from 2001 to 2010 and V.7 from 2011 to 2015), the baseline contain variables

income primarily to women, and being accessible to disadvantaged ethno-racial groups, BFP may also contribute directly to reducing race and gender, as well as socioeconomic, inequalities in CVD mortality.[44–49]

that are common to the two versions and those that are only available in one of them. Also, completeness varies widely between variables (0%–10% in the selected variables) (table 2) and over time. Our cohort also contains baseline household monthly per capita income, which was calculated to reflect BFP eligibility criteria and comprises the sum of all household members' income from work, donations, pension and other benefits in the past month of registration, divided by the number of individuals living in the household.[51] For individuals that applied during or after 2011, the work component from the monthly per capita income was calculated as the lowest value of either the total individual income from work in the past month or the sum of an individual's income from work in the past 12 months divided by 12.

Housing characteristics (such as number of rooms within the dwelling, household building material, presence of running water and sanitation) also provide a direct measure of material circumstances, which is less variable over time than income. Highest educational attainment reflects socioeconomic position in early adulthood, since for most people it remains constant after the age of 25 years. As is the case for most administrative datasets, there have been a number of changes in the data collection processes used by the BFP administrators. Changes in the collected variables were harmonised between the different versions when possible but kept separate when there were substantial differences between the studied years. Harmonisation was performed by the Cidacs/Fiocruz team when creating the 100 Million Brazilian Cohort and documentation is available online.[52]

**Table 2** Key variables available for the analysis from the 100 Million Brazilian Cohort baseline, from Bolsa Familia dataset and from mortality data

| 100 Million Brazilian Cohort baseline | |
|---|---|
| **Individual-level variables** | |
| Age (at application) | Continuous |
| Sex | Female, male |
| Relationship with the responsible person for the household | Responsible him/herself, wife/husband, son/daughter, stepson/stepdaughter, grandchildren, parent-in-law, brother/sister, son/daughter-in-law, other relative, not relative. |
| Race/ethnicity | White, brown, black, Asian, indigenous |
| Literacy | Literate or illiterate. |
| Level of education | Never went to school, preschool, literacy school, primary education (first stage—5 years), primary education (second stage—4 years), high school, higher education |
| Monthly per capita income | Sum of income from work, donations, pension and others per divided by the number of individuals in the household in the given year. |
| **Family level variables** | |
| Municipality of family home | Single identifier for every municipality |
| Region of family home | South, South-east, North, North-east, Central-West |
| Location of family home | Urban, rural |
| Housing material | Brick or cement, Taipa, Wood, Other |
| Household type | Private, improvised but private, collective, others. |
| Household water supply | Public network, Well or natural source, cistern or others |
| Sewage disposal system | Public network, Septic tank, rudimentary tank, ditch, others. |
| Electricity | Home metre, community metre, irregular electricity, gas lighting, candlelight, other |
| Waste collection | Public collection system, burned, buried, outdoor disposal, other |
| Number of individuals in the household | Continuous |
| Number of rooms in the household (including bathrooms, living room and kitchen if separated by walls) | Continuous |
| **Bolsa Familia Program variables** | |
| Benefit starting date | Date |
| Benefit ending date | Date |
| Duration of the benefit receipt | Time in years |
| **Mortality system information** | |
| Date of death | Date |

Continued

**Table 2** Continued

| 100 Million Brazilian Cohort baseline | |
|---|---|
| Place of death | Hospital, other health establishments, household, street, others |
| Municipality of death | Single identifier for every municipality |
| Medical assistance | Yes or No |
| Necropsy investigation | Yes or No |
| Main cause of death | ICD-10 categories |

ICD, International Classification of Diseases.

The extreme poverty and poverty thresholds for BFP eligibility have changed over time at a similar rate (eg, from 2004 to 2007, extreme poverty criteria changed from ≤50BRL to ≤60BRL and poverty criteria changed from ≤100BRL to≤120BRL, corresponding to a 20% increase between 2004 and 2007; table 1). To account for these changes over time, we will standardise the monthly per capita household income to the 2014 threshold so that we can use a single cut-off value in the analysis for all years. We will multiply the household per capita income value by 1.54 if individuals apply to BFP prior to 28 December 2007, by 1.283 if individuals apply to BFP between 28 December 2007 and 1 September 2009, and by 1.1 if they apply between 1 September 2009 and 31 May 2014 (see table 1, column 5).

### Bolsa Familia Program
The BFP data will provide information on the date each family member started and finished receiving BFP benefit from 1 January 2004 to 31 December 2015. For each individual, we will include information on the first and last date they received BFP benefit.

### Mortality information system
Deaths within Brazil are subject to certification by medical professionals so the causes of death (using International Classification of Diseases (ICD)-10 codes) can be ascertained reasonably precisely. Despite the significant and continuous improvement of data quality over time, regional disparities remain with the worst quality in the poorest regions and those with worse healthcare.[53 54] In 2000, 14.3% of all deaths corresponded to ill-defined causes and this proportion varied from 28.4% in the Northeast to 6.3% in the South. Since 2005, the Ministry of Health has initiated several actions aimed at improving the quality of mortality information with an emphasis on the North and Northeast regions. By 2010, the proportion of ill-defined causes had dropped to 8.6%, but regional disparities remain.[55]

### Data analysis plan
#### Data cleaning and preparation
The 100 Million Brazilian Cohort was cleaned and the variables standardised according to strict protocols

**Table 3** All-cause and cardiovascular mortality rates among individuals from the 100 Million Brazilian Cohort (N=92 million)

| | | Mean yearly mortality rates per 100 000 person years at risk during the study period | | |
|---|---|---|---|---|
| | Deaths (×10³) | Overall | Male | Female |
| **Overall population (N=92 millions)** | | | | |
| All-cause mortality rate | 1810 | 286 | 370 | 199 |
| **Cardiovascular mortality** | | | | |
| All cardiovascular mortality rate (I00-99) | 432 | 68 | 81 | 56 |
| Cerebrovascular mortality rate (I60-69) | 113 | 18 | 20 | 15 |
| Ischaemic heart diseases mortality rate (I20-25) | 117 | 19 | 24 | 13 |
| **Individuals aged 30–69 years (N=34 millions)** | | | | |
| All-cause premature mortality rate | 943 | 482 | 604 | 344 |
| **Premature cardiovascular mortality** | | | | |
| Cardiovascular disease (CVD) premature mortality rate (I00-99) | 243 | 124 | 147 | 98 |
| Cerebrovascular premature mortality rate (I60-69) | 63 | 32 | 36 | 28 |
| Ischaemic heart diseases premature mortality rate (I20-25) | 76 | 39 | 50 | 26 |

developed by Cidacs/Fiocruz. We will start by checking the linkage quality between the 100 Million Brazilian cohort and mortality, check data quality for obvious errors by inspecting the distribution of variables, will investigate the presence of missing data, and make comparisons across similar variables to identify inconsistencies. We will also explicitly check for inconsistencies in data quality between geographical regions and over time. In the presence of substantial discrepancies between regions in the ascertainment of recording of cause of deaths, we shall stratify the analyses by region. Following initial checking and further data cleaning if necessary, we will develop derived variables harmonised over time and will maintain a detailed record and reproducible syntax for the process.

### Definitions of exposures and outcomes

As BFP was first implemented in 2004, we will only include in our analysis individuals who registered to the 100 Million Brazilian Cohort baseline on the 1 January 2004 onwards (figure 2, table 3). Since many applicants will go on to receive BFP eventually, we will allocate exposure status based on whether they are a recipient within different time intervals following first application (figure 3). Approaches to classifying exposure status will include BFP recipients being defined as those who start receiving the benefit within a given amount of time after registering in the 100 Million Brazilian Cohort baseline. As BFP receipt may vary over time, we will define exposure

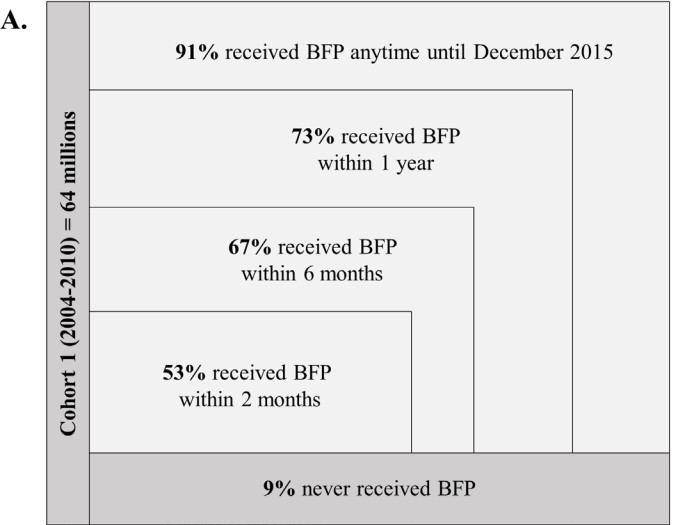
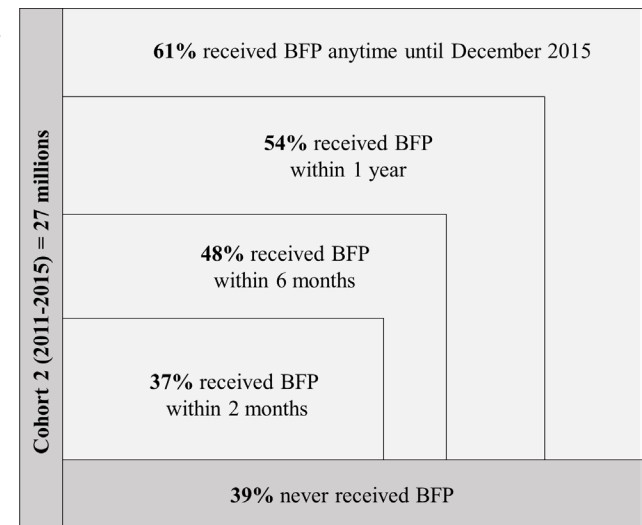

**Figure 3** Approximate cumulative number and proportion of individuals of all ages receiving Bolsa Familia Program (BFP) over time after start of follow-up for those that apply between 2004 and 2010 (cohort 1) (A) and for those that apply between 2011 and 2015 (cohort 2) (B).

status on the basis of first receipt. Once a household starts receiving BFP, individuals within the household will continue to be allocated to the exposed group (irrespective of subsequent changes to BFP status).

To evaluate all-cause and cardiovascular mortality among individuals for all ages, the follow-up time (in years) for each individual will start in different time sets after they first apply to 100 Million Brazilian cohort (ie, after the definition of individuals according to exposure status) and will end at death by any cause or on 31 December 2015. To evaluate all-cause and cardiovascular premature death, follow-up time (in years) for each individual will start when they first apply to the 100 Million Brazilian cohort or, for individuals below 30 years old, on the date they complete 30 years of age. For this analysis, follow-up time will end at death, end of follow-up (31 December 2015), or on reaching 70 years of age.

Our primary outcome is premature cardiovascular disease mortality among adults (ICD codes I00–I99), defined as deaths occurring between 30 and 69 years of age. We shall also estimate models for all-ages and premature all-cause mortality to check the sensitivity of our overall approach to the analysis, and for subgroups of cardiovascular death, including ischaemic heart disease (I20–I25), and cerebrovascular disease (I60–I69; see figure 2).

## Analysis

The final choice of methods for identifying the impact of BFP on CVD mortality will depend on initial exploratory analyses of the exposure and covariate data available in the 100 Million Brazilian Cohort. We will stratify the analyses by year of application (<2010 and ≥2011) to reflect the changes in the income calculation in both periods, which is the main eligibility criteria for BFP receipt.

For individuals that apply ≥2011, for which income data are higher quality (ie, preliminary data cleaning showed that >75% of individuals had a monthly per capita income <BRL1/USD0.25 prior to 2011), we can use a regression discontinuity design (RDD). RDD uses a threshold in a continuous ('forcing') variable that determines exposure to an intervention to distinguish exposed and unexposed units. The key identifying assumption is that units within a narrow range of values ('bandwidth') of this forcing variable either side of the threshold value will be similar in characteristics other than exposure, so that differences in outcomes can be interpreted as effects of the intervention.[56]

Standardised monthly per capita household income will be used as the forcing variable, and we will conduct the analysis for only for the poverty threshold (≤BRL154/US$39 in 2014), where we have indicative of discontinuity in the probability of receiving BFP given the monthly per capita income. Because eligibility for BFP depends partially on factors other than income, the probability of receipt will not change from 0 to 1 at the income threshold. We shall apply a fuzzy RDD, using two stage least squares, to model receipt of BFP as a function of income and the threshold in the first stage, and fitting the modelled probabilities of receipt as predictors in the outcome models.

We will estimate the effect of BFP on each of the study outcomes using Poisson or Cox proportional hazards survival models, accounting for the length of follow-up of each individual in the cohort.[45 57] Since individuals are nested within a household, we will use multilevel models to account for the lack of independence of observations.[45] In addition, to test for differences in the effect of BFP by subgroup, we will stratify the analyses by geographical region (South, Southeast, Central-west, North and Northeast) and gender (male and female), household structure, highest educational attainment, housing conditions, urbanicity and municipality level deprivation measures.

One important problem with validity in studies using RDD is manipulation of assignment, for example, by applicants understating their income. We shall plot histograms of income to test for smoothness at the thresholds, and scatterplots of covariates against income to check for any bunching below the thresholds that might indicate manipulation, and apply the McCrary test if there is any visual indication of manipulation.[58]

We shall perform a series of robustness checks and sensitivity analyses.[59] We shall use plots of the outcomes against income to decide the most appropriate functional form. Given the large numbers available for analysis in the 100 Million Brazilian Cohort, we expect to be able to restrict the analyses to households within a narrow range of income either side of the threshold, and to estimate models within a range of bandwidths to test the robustness of the assumption of local linearity, to potentially increase the generalisability of our findings and to compare with further analysis including propensity score based methods.

To improve the robustness of our results, we will perform additional analyses: (1) restricted to a subgroup of individuals whose treatment has not varied over time (ie, excluding those who stopped receiving BFP treatment); (2) restricting the follow-up time to shorter periods in which socioeconomic conditions are less likely to have varied over time; (3) exploring the possibility of treatment contamination occuring when untreated individuals start receiving the BFP, and (4) removing families with zero income or restricting the analysis to individuals that are more likely to receive the treatment (eg, monthly per capita income below a certain threshold). To test if the effect of BFP on all-cause mortality is independent of BFP's effect on homicide and other external causes of death,[60 61] we shall re-estimate the effect excluding external causes of death.

In addition, we will test the robustness of our local effect estimates by adjusting the survival models for age or other strong potential confounders. We shall also repeat the analyses in geographical areas with more and less reliable mortality data, for example, with different proportions of ill-defined causes of death and underreported mortality.

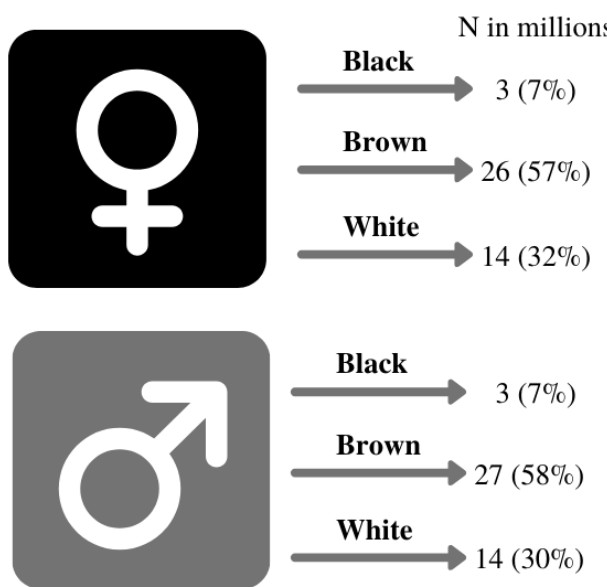

N in millions

**Black** → 3 (7%)
**Brown** → 26 (57%)
**White** → 14 (32%)

**Black** → 3 (7%)
**Brown** → 27 (58%)
**White** → 14 (30%)

**Figure 4** Main strata to investigate intersectionality regarding gender and race/ethnicity among individuals from all ages from the 100 Million Brazilian Cohort (2004–2015).

Aside from RDD, we will estimate the effect of BFP on all-cause and cardiovascular mortality using propensity score-based methods. Propensity score methods seek to reduce confounding through the comparison of units that have similar probabilities to be exposed or unexposed to the intervention, given observed characteristics.[62] We will use risk-set matching to calculate the probability of receiving BFP over time, given the baseline demographic and socioeconomic variables of the 100 Million Brazilian Cohort.[63 64] We will match BFP and non-BFP participants overall or within strata using the propensity score. Matching methods will include nearest neighbour matching using adequate calliper and replacement. We will estimate the average treatment effect (ATE) on the treated (ATT) of BFP on overall mortality and cardiovascular outcomes using Cox or Poisson models. Alternative approaches to deal with changes in BFP status over time will include truncating the follow-up period to a certain number of years, excluding households that do not receive BFP for a minimum number of years and considering the treatment indicator as varying over time.

To verify the hypothesis that matching could be pruning similar individuals, we will estimate in the unmatched cohort the ATT and the ATE using survival models weighted by the inverse of the probability of receiving the treatment (inverse probability of the treatment weighting). Also, we will estimate both ATT and ATE in quintiles or deciles of propensity score strata. For all the methods that rely on observable covariates, we will check balance of covariates between the intervention and control groups.

To deal with missing data, we will start by exploring the missingness pattern of covariates over time in our study population. Given the size of our sample and the complexity of causal inference methods, we are unable to implement

**Table 4** Time schedule for evaluation the effect of Bolsa Familia Program on all-cause and cardiovascular diseases mortality in the 100 Million Brazilian Cohort

| | 2020 | | | | | | | | | | | | 2021 | | | | |
|---|---|---|---|---|---|---|---|---|---|---|---|---|---|---|---|---|---|
| | J | F | M | A | M | J | J | A | S | O | N | D | J | F | M | A | M |
| Elaboration of the study protocol—submitted 22 April 2020 | X | X | X | X | | | | | | | | | | | | | |
| Descriptive data analysis | | | | | X | X | X | X | X | X | | | | | | | |
| Impact of Bolsa Familia on all-cause and cardiovascular mortality (data analyis) | | | | | | X | X | X | X | X | X | X | | | | | |
| Impact of Bolsa Familia on all-cause and cardiovascular mortality (writing) | | | | | | | | | | | X | X | | | | | |
| Impact of Bolsa Familia on all-cause and cardiovascular mortality with focus on gender, race and social inequality (data analysis) | | | | | | | | | | X | X | X | X | X | X | | |
| Impact of Bolsa Familia on all-cause and cardiovascular mortality with focus on gender, race and social inequality (writing) | | | | | | | | | | | | | | | X | X | X |
| Impact of Bolsa Familia all-cause and cardiovascular mortality with focus on deprivation index/inequalities (data analysis) | | | | | | | | | | | X | X | X | X | X | | |
| Impact of Bolsa Familia all-cause and cardiovascular mortality with focus on deprivation index/inequalities (writing) | | | | | | | | | | | | | | | X | X | X |

multiple imputation. For the development of the propensity score, we will try to limit inclusion of covariates to those with a relatively low percentage of missing values (eg,<5%). For variables which have higher levels of missingness but which are strongly informative of intervention receipt, we will include a missing indicator for that variable. In addition, we will perform a sensitivity analysis using only individuals without missing data in the covariates of interest (ie, complete case analysis).

Our preference, as informed by the MRC's guidance on the evaluation of natural experiments,[45] is to use both RDD and propensity score based methods, since each method relies on different assumptions (eg, no unobserved confounding for propensity score-based methods and as-if random allocation at the eligibility cut-off for BFP in the case of RDD). Obtaining consistent results from the two approaches for individuals that applied ≥2011 will, along with the other robustness checks, strengthen confidence in our inferences about impact. For each of the analyses, we will evaluate whether the assumptions of the corresponding methodology are tenable. The exploratory analyses and decisions leading to the final specification of the analyses will be fully documented and reported alongside the findings.

### Effect modification and intersectionality

We will investigate differential impacts of BFP on mortality by including interaction terms within the models and by stratifying the analyses to provide estimates for population subgroups defined by gender, race/ethnicity, urbanicity and socioeconomic position (education and area-level deprivation). To investigate intersectionality, we will test for effect measure modification on both an additive scale and a multiplicative scale.[65 66] We are especially interested in exploring variation according to combinations of characteristics, such as gender and race/ethnicity (figure 4).[65] Therefore, we will create a single categorical variable that incorporates the two concepts of interest and estimate the relative excess risk due to interaction.[66] To study intersectionality on a multiplicative scale, we will fit interaction terms between variables representing the two concepts. For further subgroup analyses, we shall submit a full specification of the analysis of interest to the Social Policy and Health Inequality (SPHI) project steering group before data analysis begins and report the results of all analyses rather than selecting according to size or significance of effects.

### Time schedule

The time schedule for the analysis is described in table 4.

### Patient and public involvement

Patients and the public were not involved in this study.

### ETHICS AND DISSEMINATION

The 100 Million Brazilian Cohort study was approved by the ethics committee of Gonçalo Muniz Institute—Oswaldo Cruz Foundation (1.612.302) and the specific aims of this project were submitted for ethical approval to the same ethics committee. The University of Glasgow Medical, Veterinary & Life Sciences College Ethics Committee also approved the study (project number: 200190001). All data are linked in a safe room with access to restricted people only. After data are linked and linkage accuracy is calculated, researchers will have full access to the deidentified dataset. The dataset will be accessed by researchers on application to a data curation committee with a detailed analysis plan. The dataset will receive a Digital Object Identifier System, and full specification of how the dataset was created will be available online. All manuscripts will be published in open-access journals. The study is part of a larger project, the National Institute for Health Research Global Health Research Group on Social Policy and Health Inequalities, which supports a communications group responsible for producing summaries of the published research results for managers, policy-makers and the broader public.

**Author affiliations**

¹Centro de Integração de Dados e Conhecimentos para Saúde (Cidacs), Fundação Oswaldo Cruz, Salvador, Brazil
²Department of Infectious Disease Epidemiology, London School of Hygiene & Tropical Medicine, London, UK
³MRC/CSO Social & Public Health Sciences Unit, University of Glasgow, Glasgow, UK
⁴Instituto de Matemática e Estatística, Universidade Federal da Bahia, Salvador, Brazil
⁵Department of Non-communicable Disease Epidemiology, London School of Hygiene & Tropical Medicine, London, UK
⁶Health Data Research (HDR), London, UK
⁷Instituto de Saúde Coletiva, Universidade Federal da Bahia, Salvador, Brazil

**Contributors** JMP, PC, EMLA and SVK wrote the first draft of the protocol. MA, LA, SA, LS, MLB and AHL contributed with additional material. All authors participated with the discussions during the development of this protocol and reviewed the final version.

**Funding** This research was funded by the National Institute for Health Research (NIHR) (GHRG /16/137/99) using UK aid from the UK Government to support global health research. The views expressed in this publication are those of the author(s) and not necessarily those of the NIHR or the UK Department of Health and Social Care. The Social and Public Health Sciences Unit is core funded by the Medical Research Council (MC_UU_12017/13) and the Scottish Government Chief Scientist Office (SPHSU13). SVK is funded by a NHS Research Scotland Senior Clinical Fellowship (SCAF/15/02). Cidacs/Fiocruz is supported by grants from CNPq/MS/Gates Foundation (401739/2015-5) and the Wellcome Trust, UK (202912/Z/16/Z).

**Competing interests** None declared.

**Patient and public involvement** Patients and/or the public were not involved in the design, or conduct, or reporting, or dissemination plans of this research.

**Patient consent for publication** Not required.

**Provenance and peer review** Not commissioned; externally peer reviewed.

**ORCID iDs**
Julia M Pescarini http://orcid.org/0000-0001-8711-9589
Srinivasa Vittal Katikireddi http://orcid.org/0000-0001-6593-9092

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
