## [Reviewer comments · BMJ Open]

ARTICLE DETAILS

TITLE (PROVISIONAL)	Evaluating the impact of the Bolsa Familia conditional cash transfer program on premature cardiovascular and all-cause mortality using the 100 million Brazilian cohort: A natural experiment study protocol
AUTHORS	Pescarini, Julia; Craig, Peter; Allik, Mirjam; Amorim, Leila; Ali, Sanni; Smeeth, Liam; Barreto, Mauricio; Leyland, Alastair; Aquino, Estela M. L.; Katikireddi, Srinivasa

VERSION 1 – REVIEW

REVIEWER	Jidong Sung Sungkyunkwan University School of Medicine Seoul, Republic of Korea
REVIEW RETURNED	20-May-2020

GENERAL COMMENTS	This is a study protocol to investigate the health effect (mainly on CVD) of The Bolsa Familia Program (BFP), a very large Brazilian conditional cash transfer (CCT) programmes. The study has a definite strength in the sample size. The protocol was carefully elaborated. Several points should be addressed before publication. 1. In Logic model in the page 6 and corresponding Figure 1, the authors suggested improved health care utilization as one of the mediators for favorable outcome, but presenting only 'controlled blood pressure' in Figure 2 seems to be too specific. There are many other preventive measures such as statins, antiplatelet agents and/or smoking cessation other than that. Consider a more comprehensive description.2. In the page 5: 'If individuals do no longer meet the inclusion criteria, ie., if they improve their socioeconomic status, do not meet the conditionalities or do not update the registry every two years, the benefit will only continue for two more years.' So, there is significant heterogeneity among those who were dropped from the CCT. Those whose SES improved are likely to have favorable health outcome but the others may have poorer outcome. Please describe the analysis plan to address this issue.3. In the page 7: 'Mortality Information System' section described possible inaccuracy problem in the outcome (mortality). Do you have any plan to test the validity of the cause of death data? If it is not possible, do you have any previous study result to estimate the validity of the cause of death in death certificates.4. In the page 8: 'For individuals that apply ≥ 2011, in which we have better recorded income data'. Does this mean that classification in the ≤ 2010 cohort may be less accurate? Please give some more detail in the rationale for this division.
---

REVIEWER	Jesús-Adrián Alvarez Interdisciplinary Centre on Population Dynamics, University of Southern Denmark
REVIEW RETURNED	24-May-2020

GENERAL COMMENTS	The study protocol describes the set up to investigate the impact of the conditional cash transfer program (Bolsa Familia Program, further referred to as BFP) on cardiovascular and all-cause mortality in Brazil. The methods to be used in the study are sound and the framework seems to be adequate. I have some minor comments.  1. The authors define premature mortality as deaths among persons 30 to 69. Why would they choose such range? Could you elaborate more? 2. Brazil depicts a large component of external mortality driven by homicides (see Alvarez et al, 2020 in Population Studies). This component is mainly observed in males between ages 15-55. It is important that the authors consider the effect of the BFP on the external mortality component, specifically when analysing the third and fifth objectives of their research protocol: "To estimate the causal effect of BFP on all-cause premature mortality." and "To explore how combinations of selected social characteristics influence the causal effects of BFP on the above outcomes, adopting an intersectionality perspective." 3. In the description of the socioeconomic variables (section datasets), the authors mention that completeness varies from variable to variable.  a. Is it possible to provide an estimate of the percentage of missing values in such variables? b. What type of missing data is? MAR, MNAR, MCAR? c. Will the data be imputed or what is the strategy to deal with such missing data? 4. When describing how to address the changes over time in eligibility criteria for BFP, the authors mention that "To account for changes in the eligibility criteria for BFP over time (Table 2), we will standardize the monthly per capita household income to the 2014 threshold. We will multiply the household per capita income value by 1.54 if individuals apply to BFP prior to 28th December 2007, by 1.283 if individuals apply to BFP between 28th December 2007 and 1st September 2009, and by 1.1 if they apply between 1st September 2009 and 31st May 2014." Could you provide more details about the why such multiplication factors? 5. When calculating exposures based on their 100 Million Brazilian cohort, it is very important that the authors take into account the different observational schemes (censoring, truncation, etc.). Such schemes could have a huge impact on their results. 6. Please provide dates (time schedule) of planned studies.
--

VERSION 1 – AUTHOR RESPONSE

RESPONSES:

Reviewer: 1 (Jidong Sung, Sungkyunkwan University School of Medicine, Seoul, Republic of Korea)

This is a study protocol to investigate the health effect (mainly on CVD) of The Bolsa Familia Program (BFP), a very large Brazilian conditional cash transfer (CCT) programmes. The study has a definite strength in the sample size. The protocol was carefully elaborated. Several points should be addressed before publication.

1. *In Logic model in the page 6 and corresponding Figure 1, the authors suggested improved health care utilization as one of the mediators for favorable outcome, but presenting only 'controlled blood pressure' in Figure 2 seems to be too specific. There are many other preventive measures such as statins, antiplatelet agents and/or smoking cessation other than that. Consider a more comprehensive description.*

R: In the logic model (in the figure and in the text) we have focused on known mechanisms in which CCTs and/or BFP may reduce cardiovascular mortality. We have also updated figure 1 to include controlled blood pressure as one example of how improved medical treatment can impact on cardiovascular mortality. In addition, given hypertension drugs are freely available through Brazil's Universal healthcare System (SUS) we also included in our model the following hypothesis: "We hypothesize that inclusion of BFP beneficiary families in the Family Health Program might promote early CVD diagnosis and better care (Rasella et al., 2014), even though Brazil has a Universal Healthcare System (SUS) and access to free hypertension drugs has substantially increased over recent decades (Emmerick et al., 2015)." (Please see section "Methods and analysis/Logic model", page 5).

2. *In the page 5: 'If individuals do no longer meet the inclusion criteria, ie., if they improve their socioeconomic status, do not meet the conditionalities or do not update the registry every two years, the benefit will only continue for two more years.' So, there is significant heterogeneity among those who were dropped from the CCT. Those whose SES improved are likely to have favorable health outcome but the others may have poorer outcome. Please describe the analysis plan to address this issue.*

R: The vast majority of people who stop receiving the intervention do so because they have improved their socioeconomic status, and not because they did not comply with BFP conditionalities. BFP works with the idea that non-compliance means the family is more vulnerable and, in these cases, they receive a visit of a social worker (Soares, 2011). Therefore, there is less heterogeneity than might be expected. It also should be noted that improved socioeconomic conditions that lead to changes in health risk behaviour are more likely to influence at short term cardiovascular disease burden, rather than mortality. We have now included the following sentence to the text: "Nevertheless, non-compliant families are thought to be more vulnerable and, in these cases, receive a visit of a social worker that will help families' compliance and their maintenance in the programme" (Please see section "Methods and analysis/Intervention", page 5/6). To address potential biases related to changes in socioeconomic and treatment status over time, we have now included more details on how to explore the impact of these potential changes: "To improve the robustness of our results, we will perform additional analyses: i. restricted to a subgroup of individuals whose treatment has not varied over time (i.e., excluding those who stopped receiving BFP treatment); ii. restricting the follow-up time to shorter periods in which socioeconomic conditions are less likely to have varied over time; iii. exploring the possibility of treatment contamination occurring when untreated individuals start receiving the BFP, [...]" (Please see section "Methods and analysis/Data analysis plan/Analysis", page 10).

- SOARES, F. (2011). Brazil's Bolsa Família: A Review. *Economic and Political Weekly*, 46(21), 55-60. Retrieved July 28, 2020, from www.jstor.org/stable/23017226

3. *In the page 7: 'Mortality Information System' section described possible inaccuracy problem in the outcome (mortality). Do you have any plan to test the validity of the cause of death data? If it is not possible, do you have any previous study result to estimate the validity of the cause of death in death certificates.*

R: We agree with the reviewer that it is necessary to evaluate possible bias due to the ascertainment of death certificates. We have now included in page 7, references that suggest that mortality records have been improved in Brazil over, despite remaining geographical inequalities: "Despite the significant and continuous improvement of data quality over time, regional disparities remain with the worst quality in the poorest regions and those with worse health care (Lima et al., 2014; Junior et al., 2017)." (Please see section "Methods and analysis/Datasets/ Mortality Information System", page 7). In addition, we also plan to stratify the analysis to geographical regions (e.g., microregions or municipalities) with better and worse quality of mortality data. Therefore, we have now rephrase the sentence in page 10 to better describe the intended analysis: "We shall also repeat the analyses in geographical areas with more and

less reliable mortality data, e.g. with different proportions of ill-defined causes of death and underreported mortality.” (Please see section “Methods and analysis/Data analysis plan/Analysis”, page 10).

4. In the page 8: ‘For individuals that apply ≥ 2011 , in which we have better recorded income data’. Does this mean that classification in the ≤ 2010 cohort may be less accurate? Please give some more detail in the rationale for this division.

R: To be more accurate in this description, we have now included additional information regarding CadÚnico registry: “The baseline of the 100 Million Brazilian Cohort includes a range of sociodemographic variables collected at individuals’ first application in CadÚnico, and includes household income, gender, age, race/ethnicity, geographical region and urban-rural classification, housing characteristics and education (Table 2). As the cohort was built from different versions of CadÚnico (ie., version 6 from 2001-2010 and version 7 from 2011-2015), the baseline contain variables that are common to the two versions and those that are only available in one of them. Also, completeness varies widely between variables (0-10% in the selected variables) (Table 2) and over time.” (Please see section “Methods and analysis/ Datasets/Sociodemographic variables”, page 6-7). For the reasons we will only use data from 2011 onwards to apply RDD, please see: “For individuals that apply ≥ 2011 , for which income data are higher quality (i.e., preliminary data cleaning showed that $>75\%$ of individuals had a monthly per capita income $< \text{BRL}1/\text{USD}0.25$ prior to 2011), we can use a regression discontinuity design (RDD).” (Please see section “Methods and analysis/Analysis”, page 9).

Reviewer: 2 (Jesús-Adrián Alvarez, Interdisciplinary Centre on Population Dynamics, University of Southern Denmark)

The study protocol describes the set up to investigate the impact of the conditional cash transfer program (Bolsa Família Program, further referred to as BFP) on cardiovascular and all-cause mortality in Brazil.

The methods to be used in the study are sound and the framework seems to be adequate. I have some minor comments.

1. *The authors define premature mortality as deaths among persons 30 to 69. Why would they choose such range? Could you elaborate more?*

R: For this study, we have used the same definition of premature mortality by Non-communicable diseases as defined in one of the Sustainable Developmental Goal (SDG) target 3.4 indicators. To clarify that in the text, we have now included the following sentence to the text: “Premature mortality (i.e., death among persons 30 to 69 years of age) is an important indicator included in the Sustainable Developmental Goals (SDG) 3.4 target for monitoring the implementation of effective public policies for disease prevention and control.” (Please see Introduction section, page 4).

2. *Brazil depicts a large component of external mortality driven by homicides (see Alvarez et al, 2020 in Population Studies). This component is mainly observed in males between ages 15-55. It is important that the authors consider the effect of the BFP on the external mortality component, specifically when analysing the third and fifth objectives of their research protocol: “To estimate the causal effect of BFP on all-cause premature mortality.” and “To explore how combinations of selected social characteristics influence the causal effects of BFP on the above outcomes, adopting an intersectionality perspective.”*

R: We agree with the reviewer and, considering that previous studies have suggested an indirect effect of Bolsa Familia program in reducing violence related deaths (homicides), we have now included the following sensitivity analysis: “To test if the effect of BFP on all-cause mortality is independent of BFP’s effect on homicide and other external causes of death (Alvarez et al, 2020; Machado et al 2018), we shall re-estimate the effect excluding external causes of death.” (Please see section “Methods and analysis/Data analysis plan/Analysis”, page 10). We note that the issue of impacts on external mortality is of considerable policy relevance but another team has been focusing on this issue using these data and we are therefore not duplicating that ongoing work.

3. *In the description of the socioeconomic variables (section datasets), the authors mention that completeness varies from variable to variable.*

- a. *Is it possible to provide an estimate of the percentage of missing values in such variables?*
 b. *What type of missing data is? MAR, MNAR, MCAR? c. Will the data be imputed or what is the strategy to deal with such missing data?*

R: We have now provided some preliminary estimates of missing data in the “datasets” section and have now included in the “Analysis” section the methods chosen to deal with missing data: “Also, completeness varies widely between variables (0-10% in the selected variables) (Table 2) and over time.” (Please see section “Methods and analysis/ Datasets/Sociodemographic variables”, page 7); “To deal with missing data, we will start by exploring the missingness pattern of covariates over time in our study population. Given the size of our sample and the complexity of causal inference methods, we are unable to implement multiple imputation. For the development of the propensity score, we will try to limit inclusion of covariates to those with a relatively low percentage of missing values (e.g., <5%). For variables which have higher levels of missingness but which are strongly informative of intervention receipt, we will include a missing indicator for that variable. In addition, we will perform a sensitivity analysis using only individuals without missing data in the covariates of interest (i.e., complete case analysis).” (Please see section “Methods and analysis/ Datasets/Sociodemographic variables”, page 11).

4. *When describing how to address the changes over time in eligibility criteria for BFP, the authors mention that “To account for changes in the eligibility criteria for BFP over time (Table 2), we will standardize the monthly per capita household income to the 2014 threshold. We will multiply the household per capita income value by 1.54 if individuals apply to BFP prior to 28th December 2007, by 1.283 if individuals apply to BFP between 28th December 2007 and 1st September 2009, and by 1.1 if they apply between 1st September 2009 and 31st May 2014.” Could you provide more details about the why such multiplication factors?*

R: In order to make this method more comprehensible, we have now included the following sentence to the beginning of the paragraph: “The extreme poverty and poverty thresholds for BFP eligibility have changed over time at a similar rate (e.g., from 2004 to 2007, extreme poverty criteria changed from ≤50 to ≤60 and poverty criteria changed from ≤100 to ≤120, corresponding to a 20% increase between 2004 and 2007) (Table 1). To account for these changes over time, we will standardize the monthly per capita household income to the 2014 threshold so that we can use a single cut-off value in the analysis for all years.” (Please see section “Methods and analysis/ Datasets/Sociodemographic variables”, page 7; and Table 1).

5. *When calculating exposures based on their 100 Million Brazilian cohort, it is very important that the authors take into account the different observational schemes (censoring, truncation, etc.). Such schemes could have a huge impact on their results.*

R: We agree and have now included additional analysis to test the robustness of our results: “To improve the robustness of our results, we will perform additional analyses: i. restricted to a subgroup of individuals whose treatment has not varied over time (i.e., excluding those who stopped receiving BFP treatment); ii. restricting the follow-up time to shorter periods in which socioeconomic conditions are less likely to have varied over time; iii. exploring the possibility of treatment contamination occurring when untreated individuals start receiving the BFP, and iv. removing families with zero income or restricting the analysis to individuals that are more likely to receive the treatment (e.g., monthly per capita income below a certain threshold)” (Please see section “Methods and analysis/Data analysis plan/Analysis”, page 10).

6. *Please provide dates (time schedule) of planned studies.*

R: We have now provided a time schedule for the development of the study.

VERSION 2 – REVIEW

REVIEWER	Jidong Sung Sungkyunkwan University School of Medicine Seoul, Republic of Korea
REVIEW RETURNED	06-Sep-2020

GENERAL COMMENTS	Overall, the points suggested by the reviewer have been properly addressed, though it seems that the authors do not describe any plan to directly confirm the validity of the cause of death data. I assume that this is not feasible in their research condition. Instead they suggest alternative analysis plan (page 10, line 44-50), which seems to be acceptable at the moment.
--

REVIEWER	Jesús-Adrián Alvarez Interdisciplinary Centre on Population Dynamics, University of Southern Denmark
REVIEW RETURNED	28-Sep-2020

GENERAL COMMENTS	The authors have fulfilled my queries about the previous version of the protocol. All the points raised were clarified.
---